# Vegan Diet and the Gut Microbiota Composition in Healthy Adults

**DOI:** 10.3390/nu13072402

**Published:** 2021-07-13

**Authors:** Emily A. Losno, Katharina Sieferle, Federico J. Armando Perez-Cueto, Christian Ritz

**Affiliations:** 1Department of Nutrition, Exercise and Sports, Faculty of Science, University of Copenhagen, Rolighedsvej 26, 1958 Frederiksberg C, Denmark; ritz@sdu.dk; 2Department of Food Science, Faculty of Science, University of Copenhagen, Rolighedsvej 26, 1958 Frederiksberg C, Denmark; apce@food.ku.dk; 3National Institute of Public Health, University of Southern Denmark, Studiestræde 6, 1455 Copenhagen K, Denmark

**Keywords:** gut microbiota, vegan diet, plant-based diet, gut bacteria, systematic review

## Abstract

The human gut microbiota are the microorganisms (generally bacteria and archaea) that live in the digestive tracts of humans. Due to their numerous functions, the gut microbiota can be considered a virtual organ of the body, playing a pivotal role in health maintenance. Dietary habits contribute to gut microbiota composition, and evidence from observational and intervention studies suggest that vegan diets may promote health, potentially through affecting the diverse ecosystem of beneficial bacteria in the gut. A systematic literature search was conducted on PubMed and Scopus to identify studies investigating the microbiota composition in vegans. Vegans are defined as people excluding food products that are derived from animals from their diet. Nine observational studies were identified. The main outcome of the systematic review was an increase in Bacteroidetes on the phylum level and a higher abundance of *Prevotella* on the genus level. In conclusion, the present systematic literature review highlighted some benefits of a vegan diet but also demonstrated the complexity of evaluating results from gut microbiota research. The available evidence only consisted of cross-sectional studies, therefore suggesting the need for well-designed randomised controlled trials. Furthermore, the quality assessment of the studies included in the review suggested a lack of standardised and validated methods for participant selection as well as for faecal sampling and faecal analysis.

## 1. Introduction

The human gut microbiota is the ensemble of all the microorganisms (mostly bacteria) that inhabit the gastrointestinal tract. It can be considered a virtual organ of the body, due to its numerous functions, such as nutrient metabolism, drug metabolism, and collaborating with our immune system to fight the colonisation of pathogenic microorganisms. The gut microbiota plays an important role in the fermentation of non-digestible substrates such as dietary fibres and endogenous intestinal mucus, which in turn produces short-chain fatty acids (SCFAs), which are important signalling molecules involved in the regulation of metabolism, inflammation, and disease [1,2]. The core microbiota consists mainly of two phyla: Bacteroidetes, which include both the genera *Bacteroides* and *Prevotella*, and Firmicutes, which include the genera *Clostridium*, *Enterococcus*, *Lactobacillus*, and *Faecalibacterium* [3]. In a cross-sectional study by Schwiertz et al., Firmicutes and Bacteroidetes were shown to make up about 95–97.7% of the total gut microbiota in vegans and omnivores, with Firmicutes contributing around 56–58.6% and Bacteroidetes contributing 39% to the total gut microbiota [3].

The diversity and functions of the gut microbiota are influenced by many different factors, including age, use of antibiotics, consumption of pro- or prebiotics, and also by dietary habits [4]. Observational studies in healthy subjects have shown that there are differences in the microbiota composition between omnivores, vegetarians, and vegans [5]. Vegetarian diets include a variety of dietary patterns, which all exclude meat, poultry, and similar products from the diet. Vegan diets, on the other hand, exclude all foods of animal origin, such as meat, fish, dairy, and eggs, and they focus mainly on fruits and vegetables, grains, legumes, nuts, and seeds [6]. Compared to omnivorous diets, vegan diets are commonly richer in fibres and lower in saturated fat and proteins. Long-term vegetable consumption has been proposed to correlate with the diversity of the gut microbiota and a higher fibre intake increased the prevalence of microbes associated with a healthy gut [7]. Especially the higher content of fibres and polyphenols in vegan and vegetarian diets seems to play the biggest role in the maintenance of a more diverse ecosystem of beneficial bacteria [8]. It increases the levels of bacteria that are capable of metabolising complex carbohydrate substrates and polysaccharides to produce SCFA, which are important for the regulation of metabolism, inflammation, and disease [9]. In addition, vegan diets are linked to lower body mass index (BMI), lower low-density lipoprotein (LDL) cholesterol, lower fasting blood glucose, insulin, blood pressure, triglycerides, and overall improved cardiovascular health [10,11,12,13]. These observations support other evidence that vegan diets are likely to lower the risk of cardiovascular disease and diabetes [14,15,16,17]. However, vegan diets also contain lower amounts of vitamin B12 [18] and vitamin D and vegans consume less than is on average required of vitamin B12, vitamin D, and calcium [19]. This can lead to deficiencies in those micronutrients and potentially creates the need for supplementation [19].

Several studies have found reduced levels of pathobionts, such as Enterobacteriaceae, which are associated with inducing low-grade inflammation, in the vegan gut microbiota [8,12,20,21]. The higher polyphenol content in vegan diets was shown to increase the abundance of *Bifidobacterium* and *Lactobacillus*, which are both beneficial for cardiovascular health, due to their anti-inflammatory effect. The increased amount of fibre that is found in vegan diets compared to omnivorous diets leads to an increase in fibre-degrading bacteria, which include *Bifidobacterium*, *Prevotella*, *Bacteroides*, and *Clostridium* [8].

Microbiota research has been growing constantly in the past few years, showing how several diseases are strongly associated with dysfunctions in the gut [22] and therefore confirming how important it is for humans to eat foods that can support the development of healthy gut microbiota.

This literature review was conducted with the aim of systematically exploring the existing literature on vegan diets and the composition of the gut microbiota and consequently comparing the possible effects of a vegan diet vs. omnivorous diet on people’s health.

## 2. Materials and Methods

### 2.1. Eligibility Criteria

The PICOS tool [23] was used to select eligible studies for this review. Table 1 summarises the PICOs used for this systematic literature review. Inclusion criteria were any observational or intervention studies conducted in human adults that reported the comparison between vegan microbiota compared with an omnivore control group. Studies with modified vegan or vegetarian diets or with a pesco-vegetarian diet and studies with concomitant intervention in the comparison group (e.g., lifestyle intervention or pharmacological treatment) were excluded. Studies without a control group were excluded too. Eligible studies had to report the gut microbiota composition or abundance of specific intestinal bacteria as either a primary or secondary outcome. All studies that reported bacteria at either phylum, class, order, family, genus, and/or species level were included. Animal studies, studies in children and specific patient groups, reviews and meta-analyses, as well as book chapters were excluded. Studies were also excluded if the study was not available in English or a full text version could not be obtained.

### 2.2. Search Strategy

The protocol for this systematic literature review was submitted to PROSPERO, which is an international database for the prospective registration of systematic reviews, for registration on 10 March 2020. On 3 August 2020, the protocol was accepted on their platform without being checked for eligibility by the PROSPERO team, due to the COVID-19 pandemic. The study protocol was registered on PROSPERO under the number CRD42020172245. The literature search was conducted between 26 June 2020 and 29 June 2020 on PubMed and Scopus by the two authors (E.A.L. and K.S.) separately. The two researchers used the same search words on both databases on different days to ensure the reproducibility of the search results. The following search words have been used on both databases: (vegan OR vegan diet OR vegan food OR vegan intervention OR plant based OR plant based diet OR plant based food OR plant based intervention) AND (gut microbiota composition OR gut microbiota OR gut bacteria OR gut protozoa OR gut flora OR intestinal microbiota OR intestinal bacteria OR intestinal flora OR alpha diversity OR relative abundance OR ecological community OR faecal microbiota OR fecal microbiota OR fecal bacteria OR faecal bacteria OR gut microbiome OR intestinal microbiome).

In total, 407 papers were obtained from the literature search on PubMed and Scopus, and after removal of duplicates, 241 papers remained. All papers were checked hierarchically by title, abstract, and full text according to the above-mentioned inclusion and exclusion criteria by two authors (E.A.L. and K.S.), using the PRISMA flowchart [24]. The number of excluded studies at each step can be seen in the flowchart (Figure 1). Additional studies, obtained through non-systematic search, have been included, if they were deemed relevant by the researchers.

### 2.3. Data Extraction

The remaining eligible papers were included in the review, and the following data were extracted: author, year and country; study design; characteristics of the vegan study population (population size, gender ratio, and age); how long the participants have followed their habitual diets; method of assessment of the gut microbiota and outcome (differences in the gut microbiota composition, differences in abundance of specific bacteria, differences in alpha diversity; both significant and non-significant outcomes were included).

### 2.4. Quality Assessment

The quality of the included articles was assessed using a version of the Newcastle–Ottawa quality assessment scale, which had been adapted for cross-sectional studies by Modesti et al. [25]. An additional adaptation was made to the NOS by deleting Section 3 regarding non-respondents, because none of the included studies in this review reported the response rate, and therefore, it would not have been relevant to score this factor. The final adjusted version of the NOS used in this review can be found. The quality was evaluated based on the study sample, the methods used to measure exposure and outcome, the statistical analysis, and the reporting of statistical information. The assessment of the studies with the NOS was performed by the two first authors independently. Differences in results were discussed, and if no agreement could be achieved, a third researcher was asked to score the papers as well.

## 3. Results

In total, seven eligible studies were identified through the systematic search strategy [12,20,26,27,28,29,30]. Two additional studies were identified through a non-systematic search and subsequently included in the analysis [3,31]. One study is from Trefflich et al. and is a systematic literature review that additionally reported the results of the authors’ own small observational study [31]. This study was not found through the systematic search, since reviews were excluded. Additionally, the study from Schwiertz et al. [3] was identified via cross-reference search in the study by Trefflich et al. This study was in fact not published on any of the databases that were used in this systematic search strategy and therefore could not be found systematically. All studies were cross-sectional studies that compared two non-randomised groups, and these were published between 2011 and 2020. Five studies were conducted in Italy [20,26,27,28,29], three were conducted in Germany [3,12,31], and one was conducted in the United States of America [30]. Three studies were conducted in the same study population, using the same Italian cohort study [26,27,28], but these were still included in the analysis, since they reported complementary outcomes. Zimmer et al. included participants in the vegan group, if they had been following the diet for at least four weeks [12], whereas Wu et al. and Schwiertz et al. included participants that were vegan for at least the last six months [3,30]. For all other studies, participants had to be on a vegan diet for a minimum of one year prior to the start of the study. The included studies can be seen in Table 2.

### 3.1. Analysis of the Microbiota

In four of the included studies, only one faecal sample was taken from each participant [12,29,30,31], whereas in four studies, three samples were taken over three weeks [20,26,27,28], and in one study, four samples were taken over two weeks, and then pooled together [3]. To analyse the microbiota composition, two studies counted colony-forming units (CFUs) on agar plates [12,20], and the remaining studies used 16S rRNA gene sequencing [27,29,30,31], 16S rRNA Denaturing Gradient Gel Electrophoresis [28], or DNA sequencing [3,26].

Most studies reported outcomes of microbial abundance on genus level [12,20,27,28,29,30,31]. However, Schwiertz et al. only reported differences between vegan and omnivorous gut microbiota on the phylum level [3], and De Angelis et al. only reported differences on the family level [26]. Trefflich et al. and Losasso et al. also compared the gut microbiota on the phylum and family level [29,31] and De Filippis et al. reported differences on the phylum level as well [27]. Only Trefflich et al., Ferrocino et al., and Zimmer et al. were also assessing the differences in the abundance of certain bacterial species [12,28,31]. The most reported groups were Bacteroidetes and Firmicutes at phylum level and *Bacteroides*, *Prevotella*, and *Bifidobacterium* at genus level.

Only three of the included studies in this review assessed alpha diversity of the gut microbiota of vegans and omnivores [27,29,30].

### 3.2. Microbiota Composition

On a phylum level, Bacteroidetes were more abundant in vegans compared to omnivores, as both De Fillippis et al. and Losasso et al. reported in their studies [27,29]. In addition, De Fillippis et al. also found a higher ratio of Firmicutes to Bacteroidetes in omnivores [27]; however, no significant difference in this ratio was found by Losasso et al. [29]. None of the other studies reported differences in Bacteroidetes and Firmicutes. Schwiertz et al. reported a higher abundance of the Proteobacteria phylum in vegans and, even though not significant, a trend to a lower abundance of Actinobacteria in the vegan group [3]. No other results on a phylum level were reported.

#### 3.2.1. *Prevotella* and *Bacteroides*

*Prevotella* was the genus that was most often reported as a result in the included studies, and results were conclusive, with *Prevotella* being higher in vegans compared to omnivores [27,28,31]. Federici et al. also reported higher levels of *Prevotella* in vegans; however, this finding was not significant [20]. For *Bacteroides*, the results were contradictory. Ferrocino et al. reported higher levels of also *Bacteroides* in vegans, together with Federici et al. (not significant) [20,28], whereas Zimmer et al. reported a lower abundance of *Bacteroides* in vegans compared to omnivores [12]. Trefflich et al. also showed a lower abundance of *Bacteroides* in vegans; however, this was not significant [31]. De Fillippis et al. also showed an increased but not statistically significant higher Prevotellaceae abundance [27], and Losasso et al. also reported no significant difference in the abundance of Prevotellaceae between omnivores and vegans [29].

Other differences between omnivorous and vegan groups on a genus level included *Lachnospira*, which was significantly associated with a vegan diet according to De Angelis et al. and De Fillippis et al. [26,27]. The abundance of both *Bifidobacteria* and *Enterobacteria* seems to be lower in the vegan population, as shown by Zimmer et al., and mentioned by Federici et al., where the difference was not significant though [12,20]. Ferrocino et al. also found a significant lower abundance of *Bifidobacteria* in vegans [28]. *Ruminococcus* and the family Ruminococcaceae were positively associated with an omnivorous diet, according to both De Angelis et al. and De Fillippis et al. [26,27]. Additionally, other bacteria that were significantly lower in the vegan population included *Streptococcus* [27], *Staphylococcus* and *Corynebacteria* [20], *Lachnoclostridium* and *Dialister invisus* [31], and lower lactic acid bacteria, including *Lactobacillus* and *Lactococcus* [3]. Significantly higher in vegans was *Roseburia* [27] and *Verrucomicrobiota* [3].

#### 3.2.2. Alpha Diversity

De Fillippis et al. and Wu et al. did not find a significant difference between vegans and omnivores [27,30], whereas Losasso et al. report a significantly lower alpha diversity in omnivores compared to the vegan group [29].

### 3.3. Quality of the Studies

The quality of the studies was assessed based on an adapted version of the Newcastle–Ottawa scale [25]. Most studies were ranked as being of medium quality. The study by De Filippis et al. received eight out of nine possible points [27], which was followed by the study by Losasso et al., which received seven points [29]. Five studies showed a medium level of quality, scoring five points out of nine [3,12,20,28,31]. Lastly, the studies by De Angelis et al. and Wu et al. received four points and, consequently, they were rated as having the lowest quality among the included studies [26,30]. In Table 3, the calculated quality scores of the included studies are shown. The score for the representativeness of the sample was equal to zero for most studies because the cohort was selected through vegan or vegetarian conventions or via platforms from vegetarian societies. Therefore, because of this recruitment method, the participants were not considered truly representative of the average in the target population. Trefflich et al. recruited participants by announcements and therefore targeted the whole population [31]. Except for De Filippis et al. and Ferrocino et al., most studies did not justify their sample size.

## 4. Discussion

Up to date, only one study has looked into the literature that compares omnivores gut microbiota and vegetarian and vegan gut microbiota [31]. In this present literature review, we have excluded the vegetarian diet from the comparison, and only a vegan gut microbiota was compared to an omnivorous control, because we believe that there are more variations in vegetarian eating patterns, and therefore, it is more difficult to interpret and compare to omnivorous diets. Moreover, due to the rising popularity of veganism and plant-based eating, focusing on vegan diets and comparing them to omnivorous diets was of higher interest to us.

The data presented were obtained from observational studies that spanned three different countries, two in Europe (Italy and Germany), and one study was conducted in the USA. The cohorts of the included studies were rather small, with a maximum of 51 participants in each group [27,28]. Statistical significance was not reached for some of the differences between the groups, which was potentially due to the small sample size and high inter-individual differences. The authors reported the gut microbiota results on various taxonomic levels, i.e., phylum, family, genus, or species level, which complicated the comparison of the findings between studies.

There were conflicting results regarding the most abundant genus in vegans vs. omnivores. Most studies supported that Bacteriodetes, and especially *Prevotella*, are more abundant in vegans compared to omnivores [20,27,28,31]. Our findings are in line with existing evidence in the literature [32,33], showing that a diet high in plant-based foods and low in animal foods is associated with an increased abundance of Bacteroidetes and that the Firmicutes/Bacteroidetes ratio is lower in vegans compared to omnivores. However, not all the included studies reported differences between vegans and omnivores on the phylum level, and therefore, the Bacteroidetes abundance was not always assessed. *Prevotella* abundance was more often reported and, as previously mentioned, *Prevotella* was associated with a vegan diet in this review, with four studies reporting higher levels in vegans [20,27,28,31]. Bacteroidetes, and especially *Prevotella*, are associated with a high intake of fibre-rich foods [34,35], which is the reason for expecting a higher abundance of these bacterial taxa in the vegan population.

There is still no definite agreement in the literature about the association of alpha diversity with dietary habits, since some studies report a higher alpha diversity in vegetarians or vegans, which is mainly facilitated by a higher intake in vegetables [7], whereas another study did not find differences in alpha diversity between people following an animal-based diet and people on a vegan diet [36]. Moreover, an animal study by Li et al. only found a relationship of diet with beta diversity but not with alpha diversity [37]. This can be seen in the results of this review as well, since only one study found significant differences in the alpha diversity between omnivores and vegans [29], with vegans having a higher diversity, and two studies not finding any differences [27,30].

The comparability of the included studies was low due to several differences in the study design and methods used. The studies from Federici et al. and De Filippis et al. used a 7-day weighed food diary, and they calculated macro- and micronutrients with a software [20,27]. Losasso et al. used a semi-quantitative food frequency questionnaire, validated with a 24 h recall protocol and also calculated nutrient intakes through a software [29]. Trefflich et al. used a three-day weighing protocol, while Wu et al. assessed the diet via a 24 h dietary recall [30,31]. The remaining studies [3,12,26,28] did not use any validated dietary assessment tools; instead, they assumed a vegan diet when participants self-reported that they did not consume animal products. To allow better comparison of cross-sectional studies, it would be helpful if future studies would use the same standardised tools to assess the dietary status of participants. Moreover, since vegan diets can differ drastically in nutrient content, the nutrient intakes of participants should be calculated with a validated protocol and reported in order to increase the quality of the study.

The comparison of human gut microbiota composition presents several challenges, some of which will be discussed in this section. Firstly, the microbiota is a constantly changing environment affected by many different factors and to different degrees. Antibiotic use, lifestyle [38], and diet, and especially the consumptions of dairy and other animal foods, have been shown to be responsible for some of the variation in the human gut microbiota [7,39]. According to Klimenko et al., a diet high in meat products and dairy is associated with a lower abundance of *Prevotella* species and higher levels of *Streptococcus*, which might be due to the higher intake of *Streptococcus* species through fermented dairy products [7]. Furthermore, age has been shown to affect the microbiota composition, with Bifidobacteriaceae decreasing with increasing age. Therefore, it is important that cohorts in observational studies have comparable ages and that the age span is not too wide [22,39]. Nevertheless, the study from Federici et al. reported differences in age between the vegan and omnivore cohort (vegans age: 33 ± 7 years, omnivores age: 41 ± 9 years); also, they did not mention any statistical adjustment for age as a confounding factor [20]. Another factor that could have affected the results is the length of the vegan diets. Six out of nine of the included studies excluded participants that followed the diet for less than 12 months. The study by Zimmer et al. included participants that had been on the vegan diet for at least 4 weeks [12], while Wu et al. and Schwiertz et al. had a minimum of 6 months as inclusion criteria [3,30]. These difference in adherence to the diet might have influenced the results. In addition, although a core gut microbiota exists on a global level [40], the geographical location where a study is performed also contributes to the gut microbiota composition [41]. In fact, results by Ferrocino et al. (2015) suggest that even in the same country, in this case Italy, the four different geographical regions where the study was performed had a higher impact on the gut microbiota than the type of diet (vegan or omnivore) [28]. Such differences between geographical locations could be related to the different consumed food items, environmental exposure, or lifestyle.

Secondly, the lack of a standardised method for faecal sampling and microbiota analysis made the comparison between different studies arduous. A few studies collected more than one stool sample [3,27,28], but the majority did not publish the microbiota composition at different time points to demonstrate the reproducibility of their results. Federici et al. was the only study to report that the microbiota was fairly stable across time, though only for their vegetarian and omnivore population [20]. On the contrary, Schwiertz et al. showed that the microbiota of one vegan participant varied substantially over time, which raises questions of whether it would be more appropriate to collect several faecal samples for observational studies [3]. The heterogeneity of the studies also poses a barrier to the classification of bacteria in terms of taxonomy and function. Not all included studies reported the microbiota on the same taxonomic level, and even different strains of the same species may have different or opposite effects [42,43]. De Angelis et al. (2020) highlighted that the phylogenetic composition was not useful to distinguish between vegans and omnivores, and therefore, analyses on the genus and species level are necessary to properly compare results [26].

### 4.1. Strengths and Limitations

Firstly, our literature review was performed in a systematic way, using two of the largest databases for scientific publications (PubMed and Scopus). Keywords were carefully selected with Boolean logic, and the literature search was performed by the two first authors separately and resulted in the same number of papers. Likewise, abstracts were also screened and resulted in the same selection of papers.

However, since two relevant studies [3,31] had not been identified through the systematic search, it is possible that the methods used for the systematic search, such as keywords and databases, could have been improved. The study from Schwiertz et al. [3] was not available on either PubMed or Scopus, suggesting that further databases should have been used in the systematic search to retrieve a higher number of relevant studies.

Another limitation of our review was that three of the included studies used the same population for their observations [26,27,28]. Nevertheless, all three studies were included, since they reported complementary results, also on different taxonomic levels, but since they used the same cohort and subsequently achieved the same results, they cannot be considered as three separate studies but more as a more detailed description of the same study.

As previously discussed, the low comparability of the study design and methods of analysis and reporting of the results of the included studies poses another limitation in this review.

Omnivore diets can differ greatly, based on the geographical regions where they are consumed, due to differences in food availability and culture. A Mediterranean diet is rich in seasonal vegetables, grains, legumes, fish, and extra virgin olive oil [44]. Likewise, a New Nordic diet contains high-quality carbohydrates such as cereals, crackers, and bread, made from whole-grain barley, oats, rye, berries, and fish as the main source of omega-3 fatty acids, similar to the Mediterranean diet [45]. Both the Mediterranean diet and the New Nordic diet differ significantly from the so-called Western diet, which is commonly consumed in the USA and large parts of Europe. The Western diet contains higher intakes of meat and meat products, refined carbohydrates, dairy and processed foods, and a lower intake of fibre [46,47,48]. There is a trend towards a higher prevalence of the Western diet, especially in lower- and medium-income countries, whereas diets high in vegetables, grains, and legumes become rarer [46]. We have not distinguished different omnivorous diets in this review. This could be a limitation as well, since for example the differences in fibre content and processed food intake in the Mediterranean diet compared to the Western diet could lead to differences in the gut microbiota composition as well, even though both diets were considered omnivorous. This could be another reason for the inconclusive results in this review. However, in the studies found through the systematic search, most did not specify the type of omnivorous diet that was used as a control group. From the studies included in this review, only one specified a Western diet as comparison [3], whereas the remaining studies only mention an omnivorous control group, making differentiation between different diet types difficult.

### 4.2. Directions for Future Research

The methodology in microbiota research should be similar across studies to facilitate the comparison of results. The reporting of several taxonomic levels and especially reporting results about the microbiota composition on the genus and species level would increase comparability [26]. Since effects of the microbiota can vary even between different species of the same genus, reporting the microbiota composition on the species level seems to be important in future research [26].

Additionally, the lack of intervention studies should be addressed by future research about the association between the gut microbiota composition and the vegan diet. Many factors influence the gut microbiota and intervention studies, in which potential confounding factors such as antibiotic use and certain lifestyle factors can be controlled for, which would allow for a better understanding of the effect a vegan diet can have on the composition of the human gut microbiota [38].

For future research, it would also be beneficial to distinguish between different types of omnivorous diets. The Mediterranean diet, the New Nordic Diet, and other regional diets vary in intake in meat and animal products, vegetables, and fibres and unprocessed foods, and especially the Western diet, which is more and more common in the USA and Europe, usually contains high amounts of processed foods, refined carbohydrates, and a low content of fibre. These possible differences in the diet make it difficult to compare between the studies.

## 5. Conclusions

A uniform and summarising conclusion was not possible due to the fairly low number of only nine retrieved studies, as well as their conflicting results. The present systematic literature review identified some potential benefits of a vegan diet. However, there is currently limited evidence to suggest that vegan diets lead to a healthier and more protective gut microbiota compared to omnivorous diets. Our research also demonstrated the complexity of evaluating results from gut microbiota research. As detailed above, the main challenges for the evaluation and comparison of studies were the high inter-individual differences, the variety of identified bacterial taxa, as well as few statistically significant differences between a vegan diet and an omnivore diet. Furthermore, the available evidence only consisted of cross-sectional studies, therefore suggesting the need for well-designed randomised controlled trials, and the quality assessment of the studies included in the review also suggested a lack of standardised and validated methods for participant selection as well as faecal sampling and faecal analysis.

## Figures and Tables

**Figure 1 nutrients-13-02402-f001:**
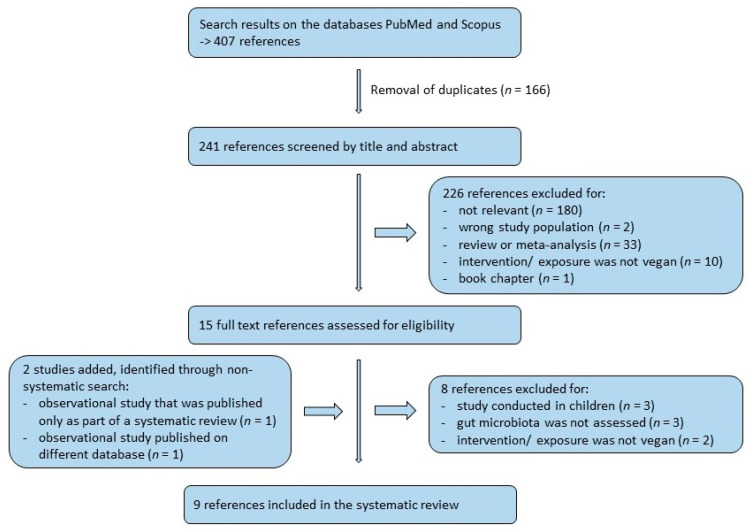
PRISMA flowchart of study selection.

**Table 1 nutrients-13-02402-t001:** PICOS for eligibility criteria.

PICOS Format	Description
Population	Presumably healthy subjects. Adult humans aged 18 years or older
Intervention or exposure	Vegan diet, defined as a plant-based diet omitting all animal products
Comparisons	Control diet (omnivore, Western-type, non-vegetarians/vegans), defined as a diet consuming any type and amount of animal products such as dairy, meat, meat products, eggs, and fish
Outcome	Gut microbiota composition through faecal samples
Study design	Cross-sectional, prospective cohort studies, randomised-controlled trials of either parallel or crossover design

**Table 2 nutrients-13-02402-t002:** Characteristics and outcome of the included studies.

Author, Country, Year	Study Design	Vegan Sample Sample Size (*n*) % Female Age (Years)	Minimum Duration of Vegan Diet (Months)	Omnivore Control Group Sample Size (*n*) % Female Age (Years)	Method of Data Collection and Gut Microbiota Assessment	Composition of Gut Microbiota	
						Significant Results	**Non-Significant Results**
De Angelis et al., Italy, 2020	Cross-sectional **	*n* = 10 (50%) 36 ± 7.0 ^§^	>12	*n* = 8	-3 samples per person collected on the same day of three consecutive weeks and then pooled together -Shotgun sequencing of the total DNA	-*Lachnospira* associated with vegans -Ruminococcaceae was the most abundant family for omnivores	
De Filippis et al., Italy, 2015	Cross-sectional *	*n* = 51 (55%) 37 ± 10	>12	*n* = 51	-3 samples per person collected on the same day of three consecutive weeks, and then pooled together -16S rRNA gene sequences	-Bacteroidetes phylum more abundant in vegans -Firmicutes: Bacteroidetes ratio was higher in omnivores -*Roseburia, Lachnospira* and *Prevotella* was lower in omnivores -*L. Ruminococcus* and *Streptococcus* correlated positively with omnivore diets and negatively with vegan diets	-alpha diversity -increased Prevotellaceae in vegans
Federici et al., Italy, 2017	Cross-sectional	*n* = 10 (30%) 33 ± 7	>12	*n* = 7 (42%) 41 ± 9	-3 samples per person, over 3 weeks; analysed individually, outcomes pooled -Bacterial counting	-*Staphylococci* and *Corynebacteria* was lower in vegans than omnivores	-lower levels of *Bifidobacteria, Enterobacteria* and mesophilic lactobacilli in vegans -*Bacteroides–Prevotella* levels being higher in vegans
Ferrocino et al., Italy, 2015	Cross-sectional *	*n* = 51 (55%) 37 ± 10	>12	*n* = 51	-Bacterial counting -16S rRNA Denaturing Gradient Gel Electrophoresis (rRNA-DGGE) -3 samples per person collected on the same day of three consecutive weeks, and then pooled together	-*Bacteroides fragilis* were higher in the omnivore group -*Bacteroides* and *Prevotella* load was higher in vegans compared to omnivores -*Coliforms* and *Bifidobacteria* lower in vegans compared to omnivores -main difference was seen when comparing the sites instead of the diet	
Losasso et al., Italy, 2018	Cross-sectional	*n* = 26 (65%) 39.4 ± 11.1	>12	*n* = 43 (73.3%) 45.0 ± 13.9	-1 sample per person but two independent total DNA extractions -16S rDNA gene	-higher counts of Bacteroidetes on vegan diet -lower alpha diversity in omnivores compared to vegans	-no difference in Firmicutes/Bacteroidetes ratio -no difference in Prevotellaceae abundance
Wu et al., USA, 2016	Cross-sectional	*n* = 15	>6	*n* = 16	-1 stool sample per person -16S rRNA gene sequences		-no taxa differed significantly in presence/abundance at genus level (after correction for multiple comparisons) -measures of diversity and evenness not sig. different between groups
Zimmer et al., Germany, 2011	Cross-sectional	*n* = 46 (39%) 46.5 ± 12.62 ^#^	>4 weeks	*n* = 46 (39%) 46.5 ± 12.26 ^#^	-1 stool sample per person -Bacterial counting	-lower *Bacteroides, Bifidobacterium* and Enterobacteriaceae in vegans	-lower *Escherichia Coli* -Significantly lower stool pH
Trefflich et al., Germany, 2019	Cross-sectional	*n* = 36 (50%) 37.5 (32.5–44.0)	>12	*n* = 36 (50%) 38.5 (32.0–46.0)	-1 stool sample per person -16S rRNA gene sequencing	-*Lachnoclostridium* and *Dialister invisus* were significantly higher in omnivores	-*Bacteroides* was higher in omnivores (21.7% vs. 17.4%) -*Prevotella* and *Faecalibacterium* was higher in vegans (10.8% vs. 7.1%)
Schwiertz et al., Germany, 2016	Cross-sectional	*n* = 9 (77.8%)	≥6	*n* = 10 (50%)	-samples from 4 time points (day 1, 4, 7, and 14) were analysed -PCR amplification, microbial DNA sequencing, detection of number of bacteria by comparison of fluorescence with standard curve of appropriate reference organism	-higher *Proteobacteria* and lower *Verrucomicrobiota* in omnivores -lower lactic acid bacteria (*Lactobacillus* and *Lactococcus*) in vegans	-lower abundance of *Actinobacteria* in vegans

Numbers are reported as mean ± SD or median (Interquartile Range (IQR)) * Same cohort and study design but different analyses of samples; ** Population selected from the same study cohort as De Filippis et al. and Ferrocino et al.; ^§^ Study population includes vegans, vegetarians, and omnivores. No information about the numbers in each group; ^#^ Only the results of a sub-group analysis are reported.

**Table 3 nutrients-13-02402-t003:** Quality assessment of included studies.

Study	Representativeness of the Sample (Max. 1 Star)	Sample Size (Max. 1 Star)	Ascertainment of the Exposure (Max. 2 Stars)	Comparability (Max. 2 Stars)	Assessment of the Outcome (Max. 2 Stars)	Statistical Test (Max. 1 Star)	Total Quality Score (Max. 9 Stars)
De Angelis et al.	/	/	/	*	**	*	4
De Filippis et al.	/	*	**	**	**	*	8
Federici et al.	/	/	**	/	**	*	5
Ferrocino et al.	/	*	/	*	**	*	5
Losasso et al.	/	/	**	**	**	*	7
Wu et al.	/	/	*	/	**	*	4
Schwiertz et al.	/	/	/	*	**	*	5
Trefflich et al.	*	/	*	**	**	/	5
Zimmer et al.	/	/	/	**	**	*	5

/: criteria were not met; *, **: 1 or 2 stars given depending on the criteria met.

## Data Availability

Not applicable.

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
