# Peer review of "Vegan Diet and the Gut Microbiota Composition in Healthy Adults"

_nutrients, 2021, doi:10.3390/nu13072402_

Round 1
Reviewer 1 Report
This is an interesting paper, which presents the lack of evidence in relation to the benefits (or not) of the vegan diet.
There are a couple of issues however, that need to be addressed for this to become more useful to the scientific community:
a) In the introduction, it is quite important to distinguish the vegan diet from others. Although many scientists (some of which in inverted commas) treat plant-based diets as a homogenous mass, they are not. I would therefore, request that the readers focus on papers that are based on the vegan diet only. It is also important to present the negative effects of the vegan diet on health and physiology (e.g., iodine/zinc deficiency which create a need for supplements etc) - this will allow for a more balanced presentation of facts. This should be noted in the abstract as well.
b) The Methodology is clearly presented with a small number of (yet important) exceptions: For example, could I assume that you included regional diets as comparators as well? If yes, do say so, because the way that you write, it is unclear (as you speak about Western diet etc). So, do give examples. You should also clarify that gut microbiota could be presented as either a primary or secondary outcome.
c) It is interesting to note, how you managed to miss 2 studies from the systematic search. Could you please explain why? The fact that you didn't miss them at the end is important yet, it is worrying because you might have missed more as well.
d) NOS is appropriate for non-randomised studies, so it might be a good idea that you used it. However, you need to specify what changed did you make in it, and explain how these changes make your findings more accurate. I could argue that scoring for some of your included studies in on the high side, so I would recommend that you have two of your researchers independently scoring them and see what you get.
e) The discussion presents and explains well the low quality of the evidence. To improve this further could I suggest that you: i) present the differences/benefits in gut microbiota that are introduced by dairy and animal products (which is a better word to use here rather than "foods" that you use), ii) be honest and suggest that the current evidence doesn't substantiate the "go-for-it" attitude that exists in regards to the suggested benefits of the vegan diet in relation to the gut microbiota iii) think again about using the 3 different papers from the same cohort, especially as they in reality they repeat the same data.
Author Response
Reviewer 1
This is an interesting paper, which presents the lack of evidence in relation to the benefits (or not) of the vegan diet.
Reply: Thanks so much for the positive feedback.
There are a couple of issues however, that need to be addressed for this to become more useful to the scientific community:
a) In the introduction, it is quite important to distinguish the vegan diet from others. Although many scientists (some of which in inverted commas) treat plant-based diets as a homogenous mass, they are not. I would therefore, request that the readers focus on papers that are based on the vegan diet only. It is also important to present the negative effects of the vegan diet on health and physiology (e.g., iodine/zinc deficiency which create a need for supplements etc) - this will allow for a more balanced presentation of facts. This should be noted in the abstract as well.
Reply: Thank you for the feedback. We agree that a plant-based diet can have different meanings to the readers, we have therefore now changed plant-based diet to vegan diet throughout the text to avoid confusion. We also added a more detailed description of what vegan means, and specified that vegans are people that do not consume any food of animal origin. We also write in the eligibility criteria that : "Studies with modified vegan or vegetarian diets or with a pesco-vegetarian diet and studies with concomitant intervention in the comparison group (e.g. lifestyle intervention or pharmacological treatment) were excluded.”, to show that only regular vegan diets were assessed.
b) The Methodology is clearly presented with a small number of (yet important) exceptions: For example, could I assume that you included regional diets as comparators as well? If yes, do say so, because the way that you write, it is unclear (as you speak about Western diet etc). So, do give examples. You should also clarify that gut microbiota could be presented as either a primary or secondary outcome.
Reply: We are not quite sure what was meant by regional diets. Western diet was only mentioned as because it is often used to describe the modern omnivore diet. Though the purpose of our study is not to differentiate between different omnivore diets, so we did not differentiate between them during our research.
Gut microbiota could be presented as either primary or secondary outcome in the included studies, thank you very much for pointing out the missing information in the methods. We have included this accordingly.
c) It is interesting to note, how you managed to miss 2 studies from the systematic search. Could you please explain why? The fact that you didn't miss them at the end is important yet, it is worrying because you might have missed more as well.
Reply: We tried to explain the reasons for not finding these studies through the systematic search in the limitations section. We do agree though, that it is relevant to already mention this in the methods and have therefore added a section about the missing studies in the methods section. We also discuss how this potentially means we have missed other studies through our research method in the limitations. Thanks for the comment.
d) NOS is appropriate for non-randomised studies, so it might be a good idea that you used it. However, you need to specify what changed did you make in it, and explain how these changes make your findings more accurate. I could argue that scoring for some of your included studies in on the high side, so I would recommend that you have two of your researchers independently scoring them and see what you get.
Reply: We have added a description of how the NOS was adapted and added both the original and the adapted version in the appendix for clarification.
The studies were scored by 2 researchers individually and we have now also included an description of the process in the methods.
e) The discussion presents and explains well the low quality of the evidence. To improve this further could I suggest that you: i) present the differences/benefits in gut microbiota that are introduced by dairy and animal products (which is a better word to use here rather than "foods" that you use), ii) be honest and suggest that the current evidence doesn't substantiate the "go-for-it" attitude that exists in regards to the suggested benefits of the vegan diet in relation to the gut microbiota iii) think again about using the 3 different papers from the same cohort, especially as they in reality they repeat the same data.
Reply: Thank you for the useful feedback.
Concerning i) in the discussion we mention some differences between omnivores and vegans in regards to dairy and meat consumption. We have now added a short descriptions about associations of meat and dairy with the gut microbiota.
However, since the main objective of this review was to research the associations of a vegan diet with the gut microbiota, we deemed this sufficient.
ii) it was not in the scope of the review to define whether vegan diets should or not be followed. Our purpose is to present the existing data on the differences reported by other studies.
iii) The 3 studies were all included since they reported complementary outcomes and therefore were all 3 deemed relevant for the results and discussion to add data and information. We therefore think it is relevant to include all 3 studies. With discussing this problem also in the limitations we hope to show that we are aware of this issue and allow the reader to draw their own conclusions.
Reviewer 2 Report
The authors undertook a systematic literature review of the effects of a vegan diet on gut microflora and compared the effects with an omnivorous diet. The article was written in good scientific language in an excellent clear way, it is good to read each part of the article. I have no comments on the English language.
The introduction provides a good introduction to the subject matter of the article and contains many relevant articles. I have comments on the Materials and Methods section:
line 92- please explain the PROSPERO name for the readers
This section should include information and explanation why 2 more publications were added to the review beyond the article classification protocol used. This is very important for the reader as well as for the methodology of the article. I think the sentences from the Discussion section of the paragraph Strengths and limitations are perfect for this purpose. Then the first two sentences of the Results section would not have introduced such confusion.
In Figure 1, the authors should consider adding mentioned 2 literature items from outside the classification protocol.
Table 3 – please add a footnote about the symbols used for the quality assessment of included studies
The Discussion section was well done, discussing aspects relevant to the subject matter of the review. However, in the conclusions, I expected more relevant information and future research directions to be given regarding the subject matter undertaken, as well as recommendations for improving the methodology of future research.
The literature cited is relevant, modern and up-to-date.
Author Response
Reviewer 2
The authors undertook a systematic literature review of the effects of a vegan diet on gut microflora and compared the effects with an omnivorous diet. The article was written in good scientific language in an excellent clear way, it is good to read each part of the article. I have no comments on the English language.
Reply: Thanks for the great feedback, much appreciated.
The introduction provides a good introduction to the subject matter of the article and contains many relevant articles. I have comments on the Materials and Methods section:
line 92- please explain the PROSPERO name for the readers
Reply: Thank you, glad you reported this missing information. We have added the explanation.
This section should include information and explanation why 2 more publications were added to the review beyond the article classification protocol used. This is very important for the reader as well as for the methodology of the article. I think the sentences from the Discussion section of the paragraph Strengths and limitations are perfect for this purpose. Then the first two sentences of the Results section would not have introduced such confusion.
Reply: We have now included a subsection both in methods and results and also discussion about the extra papers found.
In Figure 1, the authors should consider adding mentioned 2 literature items from outside the classification protocol.
Reply: Thank you! We have now modified figure 1 and added the 2 extra studies
Table 3 – please add a footnote about the symbols used for the quality assessment of included studies
Reply: Thank you, we included footnotes now.
The Discussion section was well done, discussing aspects relevant to the subject matter of the review. However, in the conclusions, I expected more relevant information and future research directions to be given regarding the subject matter undertaken, as well as recommendations for improving the methodology of future research.
Reply: Thanks for the feedback, we have now elaborated on future perspectives of gut microbiota research and how studies could increase comparability.
The literature cited is relevant, modern and up-to-date.
Reply: Thanks!
Round 2
Reviewer 1 Report
I would like to thank the authors for responding.
A couple of the points that I raised haven't been addressed so they need to be taken seriously.
a) Regional diets cover all the diets that originated in specific regions of the world. For example, the Mediterranean diet is one of them (and so the Japanese, the New Nordic etc). These cannot be treated as equal to any other Western diet, and in actual fact do affect the gut microbiota in a different way to what is defined in the manuscript as "Western diet". Therefore, the authors should distinguish between them - it would be incorrect not to do so, because clearly eating 5 days per week burgers doesn't affect our health and body the same to eating 1 day chicken, 2 days vegetables, 2 days fish etc. I would like to see better categorisations.
b) I do agree that you can't really tell anyone to follow a specific diet or not. But you can and should clear any misconception or misinterpretation of facts: therefore, you do need a much clearer statement that in contrast to popular belief the evidence is limited and more work is needed to support that vegan diets offer benefits to the gut microbiota.
Author Response
Reviewer 1 Round 2
I would like to thank the authors for responding.
A couple of the points that I raised haven't been addressed so they need to be taken seriously.
a) Regional diets cover all the diets that originated in specific regions of the world. For example, the Mediterranean diet is one of them (and so the Japanese, the New Nordic etc). These cannot be treated as equal to any other Western diet, and in actual fact do affect the gut microbiota in a different way to what is defined in the manuscript as "Western diet". Therefore, the authors should distinguish between them - it would be incorrect not to do so, because clearly eating 5 days per week burgers doesn't affect our health and body the same to eating 1 day chicken, 2 days vegetables, 2 days fish etc. I would like to see better categorisations.
Reply: Thank you very much for the clarification. We agree, that for example the Mediterranean diet and the Western diet differ in the intake in many nutrients and will also differ in the effect on the microbiota composition. Most studies did however not distinguish between different omnivorous diets and defined omnivorous diets as any diet that included animal products like meat, fish, dairy and eggs. We therefore didn’t distinguish different omnivorous diets either, since almost no data to this was reported, but stayed with the definition of an omnivorous diet including regular intake of animal products. We have, however, added a section to the discussion of the review, where we explain the differences between different regional diets and how they could possibly affect the gut microbiota as well.
b) I do agree that you can't really tell anyone to follow a specific diet or not. But you can and should clear any misconception or misinterpretation of facts: therefore, you do need a much clearer statement that in contrast to popular belief the evidence is limited and more work is needed to support that vegan diets offer benefits to the gut microbiota.
Reply: Thank you for the clarification, we agree that this has to be made clearer in the conclusion and have therefore clarified our statement in the conclusion.